

# Urine based near-infrared spectroscopy analysis reveals a noninvasive and convenient diagnosis method for cancers: a pilot study

Jing Zhu[1], Siyu Zhang[2], Ruting Wang[3,4], Ruhua Fang[1], Lan Lei[5], Ji Zheng[6] and Zhongjian Chen[3,4]

[1] Department of Clinical Laboratory, Zhejiang Cancer Hospital, Hangzhou, Zhejiang, China
[2] Department of Clinical Pharmacy, The First Affiliated Hospital, Zhejiang University School of Medicine, Hangzhou, Zhejiang, China
[3] Experimental Research Center, Zhejiang Cancer Hospital, Hangzhou, Zhejiang, China
[4] Zhejiang Key Laboratory of Diagnosis & Treatment Technology on Thoracic Oncology (Lung and Esophagus), Hangzhou, Zhejiang, China
[5] Zhejiang Hospital, Hangzhou, Zhejiang, China
[6] Department of Radiotherapy and Chemotherapy, Ningbo No. 2 Hospital, Ningbo, Zhejiang, China

Corresponding authors
Ji Zheng, 2859854552@qq.com
Zhongjian Chen, chenzj@zjcc.org.cn

## ABSTRACT

**Background**. The challenges in cancer diagnosis underline the need for continued research and development of new diagnostic tools and methods. This study aims to explore an effective, noninvasive, and convenient diagnostic tool using urine based near-infrared spectroscopy (NIRS) analysis combined with machine learning algorithm.

**Methods**. Urine samples were collected from a total of 327 participants, including 181 cancer cases and 146 healthy controls. These participants were randomly spit into train set ($n = 218$) and test set ($n = 109$). NIRS analysis ($4,000 \sim 10,000\,cm^{-1}$) was performed for each sample in both train and test sets. Five pretreatment methods, including Savitzky-Golay (SG) smoothing, multiplicative scatter correction (MSC), baseline removal (BSL) with fitting polynomials to be used as baselines, the first derivative (DERIV1), and the second derivative (DERIV2), and combination with "scaling" and "center", were investigated. Then partial least-squares (PLS) and linear support-vector machine (SVM) classification models were established, and prediction performance was evaluated in test set.

**Results**. NIRS had greatly overlapping in peaks, and PCA analysis failed in separation between cancers and healthy controls. In modeling with urine based NIRS data, PLS model showed its highest prediction accuracy of 0.780, with DERIV2, "scaling" and "center" pretreatment, while linear SVM displayed its best prediction accuracy of 0.844, with raw NIRS. With optimization in SVM, the prediction accuracy could improve to 0.862, when the top 262 features were involved as variables.

**Discussion**. This pilot study combining urine based NIRS analysis and machine learning is effective and convenient that might facilitate in cancer diagnosis, encouraging further evaluation with a large-size multi-center study.

# INTRODUCTION

Cancer is a devastating and unpredictable disease that poses a serious threat to human health and safety. The International Agency for Research on Cancer reported that there were approximately 19.3 million new cases of cancer (excluding nonmelanoma skin cancer) and 9.95 million cancer-related deaths (excluding nonmelanoma skin cancer) worldwide in 2020 (*Sung et al., 2021*). Among them, breast cancer in women has the highest incidence rate, with approximately 2.3 million new cases, followed by lung cancer. Colorectal and gastric cancers ranked third and fifth, respectively, among digestive tract tumors (*Rawla, Sunkara & Barsouk, 2019*). Lung cancer remains the leading cause of cancer death, accounting for 18%, followed by colorectal, liver, gastric, and breast cancers (*Ferlay et al., 2015*). Tragically, it is estimated that the number of cancer patients worldwide will increase by 47% to 28.4 million by 2040 compared to 2020 (*Bray et al., 2018*). With its high incidence and mortality rates, cancer poses a significant global health challenge (*Global Burden of Disease Cancer Collaboration, 2018*), about 50% of new cancer patients and 58.3% of cancer-related deaths occurred in Asia in 2020, which is related to the social and economic development of these regions and countries, as these regions and countries have insufficient or inadequate investment in cancer prevention, diagnosis, and treatment (*Ferlay et al., 2019*). Therefore, cancer diagnosis, especially early diagnosis is crucially in need.

In the present, cancer diagnosis techniques include serum markers, radiological, endoscopic techniques, and histopathological examination of tissue biopsy (as the gold standard for diagnosing cancers) (*Sung et al., 2021*). However, these diagnostic techniques have the disadvantages of low specificity, low sensitivity, time consuming, high cost, and invasiveness. Therefore, it is necessary to develop novel, economical, and effective diagnostic methods for cancers. Near-infrared spectroscopy (NIRS) is an economical technique, which is belong to molecular vibration spectroscopy and provides information of functional groups for components in samples (*Maule & Merletti, 2012*; *Litwin & Tan, 2017*). In recent years, the application of NIRS in the field of cancer diagnosis has been increasingly reported, mainly in the analysis of samples such as cells, serum, and tissues (*Allemani et al., 2018*). However, the use of urine based NIRS for cancer diagnosis has not been thoroughly investigated (*Murtaza et al., 2013*).

Urine is a fluid produced by the filtration of glomeruli, reabsorption, secretion, and excretion processes of renal tubules and collecting ducts (*Chen, Bode & Dong, 2017*). Urine contains different biological metabolites, and proteins (*Gajjar et al., 2013*), urine-based research for biomarkers has become increasingly interested. Urine, as a biological diagnostic sample type, has the advantages of convenience and non-invasive sampling, easy storage and transportation (*Murtaza et al., 2013*). In this pilot study, we mainly collected urine samples from various types of cancer patients and healthy controls, and performed NIRS

analysis and machine learning modeling, which revealed the potential diagnostic role of urine based NIRS in cancers.

## MATERIALS & METHODS

### Participants and urine sample collection

A total of 181 cancer patients as well as 146 healthy controls were recruited from Zhejiang cancer hospital, with the approval of the Ethics Committee of Zhejiang Cancer Hospital (IRB-2023-375). We received written informed consent from all the participants. Of 181 cancers, there were 62 lung cancer cases, 32 gastric cancer cases, 19 cervical cancer cases, 14 colon cancer cases, 12 thyroid cancer cases, 11 ovarian cancer cases, 10 breast cancer cases, nine liver cancer cases, nine nasopharyngeal carcinoma cases, two bladder cancer cases, and 1 kidney cancer case. The age and sex between cancer group and healthy control had no significant difference (Table 1). Before collecting a urine specimen, one should avoid vigorous exercise. In this study, morning urine was collected as the specimen, which refers to the first midstream urine sample after waking up in the morning. During collection, be sure to avoid contamination from urethral secretions, menstrual blood or vaginal secretions, semen or prostatic fluid, stool, and other substances. Use a disposable urine cup for the urine specimen, and label the container with the patient's name and unique identifier. The urine specimen should be sent for testing within 2 h after collection. Upon receiving the specimen, mix it well, draw one mL of urine, and store it at $-80\ °C$ until analysis. The study was approved by Zhejiang Cancer Hospital Committee and informed consent was received from all patients.

### NIRS data collection

Frozen urine samples were thawed at ice before NIRS analysis. The procedure for NIRS analysis was according to our recent plasma based NIRS study (*Zhu et al., 2023*). Briefly, Antaris™ II FT-NIR analyzer (Thermo Fisher Scientific, Waltham, MA, USA) was used for NIRS collection with air as the reference. A total of 200 µL urine sample was loaded in a quartz colorimetric tube with an optical path of six mm, and NIRS was generated by averaging 32 successive scans, ranging from 4,000 to 10,000 $cm^{-1}$, with a resolution of four $cm^{-1}$. The spectra were measured based on molar absorptivity, and the average spectrum was obtained for each sample by taking three measurements and processing the data using TQ Analyst 8.0 software.

### Data analysis

#### Train and test data set

The total of 327 participants were randomly divided into train set ($n = 218$) and test set ($n = 109$). The training set contained 116 cancer cases and 102 controls, while the test set had 65 cancer cases and 44 controls. The detailed number of each cancer type were listed in File S1.

#### Pretreatment of the NIRS data

To make the data more comparable and suitable for modeling, R package *hyperSpec* (version 0.100.0) was used to perform pre-treatment of NIRS raw data before machine learning

**Table 1  Basic information for the participants in this study.**

| Parameter | Cancer group (n = 181) | Healthy control (n = 146) | P-value[a] |
|---|---|---|---|
| **Age** | | | |
| >60 | 78 | 74 | 0.209 |
| <= 60 | 103 | 72 | |
| **Sex** | | | |
| Female | 82 | 71 | 0.626 |
| Male | 99 | 75 | |
| **Subgroup** | | | |
| Lung cancer | 62 | NA | |
| Gastric cancer | 32 | NA | |
| Cervical cancer | 19 | NA | |
| Colon cancer | 14 | NA | |
| Thyroid cancer | 12 | NA | |
| Ovarian cancer | 11 | NA | |
| Breast cancer | 10 | NA | |
| Liver cancer | 9 | NA | |
| Nasopharyngeal carcinoma | 9 | NA | |
| Bladder cancer | 2 | NA | |
| Kidney cancer | 1 | NA | |

Notes.

[a] Pearson's Chi-squared test was performed, and a P-value less than 0.05 was considered as significant.

(ML) modeling. Five methods, including Savitzky-Golay (SG) smoothing, multiplicative scatter correction (MSC), baseline removal (BSL) with fitting polynomials to be used as baselines, the first derivative (DERIV1), and the second derivative (DERIV2). Furthermore, pretreated NIRS data from both train and test set underwent "center" and "scale" according to means and standard deviations of NIRS data in train set.

### PLS and SVM Modeling and testing

Unsupervised principal component analysis (PCA) was initially used to detect the separation trend of the samples in train set. Then, the partial least-squares (PLS) methods were trained with the raw NIRS data and the above pretreated NIRS data from train set using R package *caret* (version 6.0-93). The support-vector machine (SVM) with linear kernel was trained using R package *e1071* (version 1.7-3; https://cran.r-project.org/web/packages/e1071/index.html) Ten repeated five-fold cross validation was performed in training. For PLS model, number of components were optimized, while for SVM model, penalty parameter C (cost) of SVM ranging from 0.1 to 20 were screened. Trained models then were used to predict the class for each unknown sample in test set, and the confusion matrix was used to calculate prediction performance. Additionally, receiver operating characteristic curve (ROC) was analyzed for each model with different pretreated NIRS data.

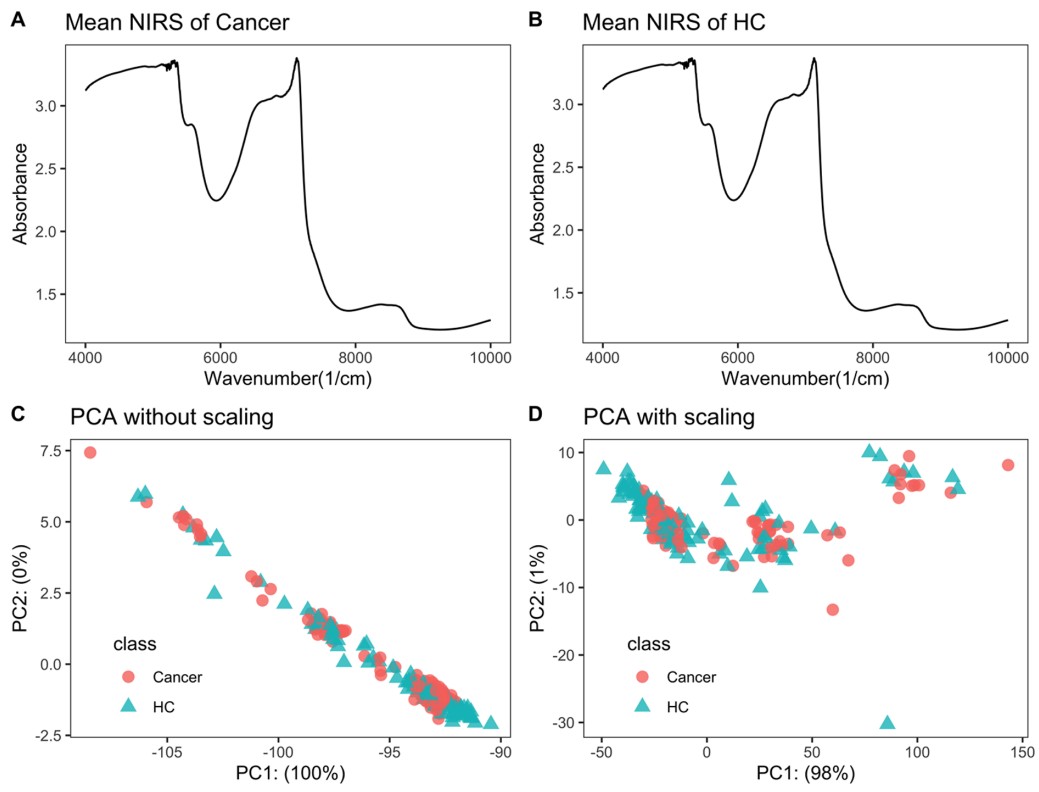

**Figure 1  NIRS PCA.** NIRS data of urine samples and its initial investigation by PCA analysis. (A) Mean NIRS of cancer patients, (B) mean NIRS of healthy controls (HC), PCA score plot using the raw NIRS data (C), and scaled NIRS data (D).

### Feature importance ranking by SVM-recursive feature elimination (SVM-RFE)

To discover the most significant NIRS features in SVM model, SVM-recursive feature elimination (SVM-RFE) algorithm was used to rank the NIRS features. The method was according to our previous studies (*Zhu et al., 2023*; *Chen et al., 2021*). Moreover, to optimize the SVM model through involving less variables, modeling progressively with different numbers of NIRS features (from Top 1 to Top N) were investigated of their prediction performance.

## RESULTS

### Raw NIRS data and pretreated NIRS data

Raw NIRS data is listed in File S2. Each NIRS spectrum consisted of 1,557 points ranging from 3,999.64 to 10,001.03 $cm^{-1}$. The average NIRS of cancers almost overlapped with that of healthy controls (Figs. 1A–1B).
## PCA analysis showed no separation trend between cancers and controls

When PCA analysis of raw NIRS data from train dataset without scaling, the first component already had 100% of the data variance, but there was no significant separation between cancers and controls (Fig. 1C). The PCA analysis of NIRS data scaling with "center" and "scale" also showed that there were greatly overlapping between cancers and control (Fig. 1D). The result implied that urine NIRS data had no direct linear relationship with their classes.

## Influence of pretreatment methods on the prediction performance of PLS and SVM

For PLS modeling with different pretreated NIRS data, the second derivative pretreatment improved the prediction accuracy for test samples, while the other methods did not influence the prediction performance or even slightly decreased the prediction accuracy, such as MSC and the second derivative, compared to modelling with raw NIRS data (Table 2 and Figs. 2A–2F). When further combined with scaling treatment, the result showed that the combinations of scaling and MSC, the first derivative, and the second derivative, evidently increased the prediction performance (Table 2 and Figs. 3A–3F). Thus, PLS modeling with NIRS after both the second derivative and scaling achieved the best prediction performance, in which the prediction accuracy in test set was 0.78, and the AUC $_{ROC}$ was 0.853). The detailed prediction performance for modeling with other treatments were demonstrated in Table 2, and Figs. 2 and 3.

For SVM modeling with single pretreatment, the result showed that SVM with the second derivative pretreatment achieved the best prediction performance, and SG smoothing had no influence on the prediction accuracy, while other methods decreased the prediction performance (Table 2, Fig. 4). In modeling with combination of scaling and other pretreatments, the result showed scaling dramatically improved the prediction performance for models. The SVM models with combinations of the first derivative pretreatment and scaling had the highest accuracy of 0.853. In terms of ROC analysis result, SVM models with combinations of raw and scaling, SG and scaling reached the highest AUC $_{ROC}$ of 0.927. Therefore, SVM with scaling treatment was the best models in this study. The detailed prediction performance for SVM modeling were illustrated in Figs. 4 and 5, and Table 2. Additionally, the cost value in SVM was optimized at 8, with which the model was able to reach its max prediction accuracy of 0.862.

In our study, SG pretreatment had almost none influence on the prediction performance for both PLS and SVM models, leading to the same prediction results and ROC curves from modeling with raw data and SG pretreated data (Table 2, Figs. 2A, 2B, 3A, 3B, 4A, 4B, and 5A, 5B). During modeling, it was observed that low specificity of PLS and SVM was the main factor resulting in a low prediction outcome. The different pretreatment methods influenced the specificities of models. For example, the first and second derivative decreased the specificities in SVM modeling, while scaling NIRS significantly increased the specificities in SVM models (Table 2). The detailed result of prediction performance in training steps were listed in File S3.

**Table 2  Prediction performance of modeling with different pre-treatment methods (testing).**

| Pretreatment[a] | Model | Sensitivity | Specificity | PPV[b] | NPV[c] | Precision | Recall | Accuracy |
|---|---|---|---|---|---|---|---|---|
| Raw | PLS | 0.754 | 0.523 | 0.700 | 0.590 | 0.700 | 0.754 | 0.661 |
| SG | PLS | 0.754 | 0.523 | 0.700 | 0.590 | 0.700 | 0.754 | 0.661 |
| MSC | PLS | 0.708 | 0.455 | 0.657 | 0.513 | 0.657 | 0.708 | 0.606 |
| BSL | PLS | 0.723 | 0.523 | 0.691 | 0.561 | 0.691 | 0.723 | 0.642 |
| DERIV1 | PLS | 0.692 | 0.455 | 0.652 | 0.500 | 0.652 | 0.692 | 0.596 |
| DERIV2 | PLS | 0.831 | 0.591 | 0.750 | 0.703 | 0.750 | 0.831 | 0.734 |
| Raw | SVM | 0.708 | 0.636 | 0.742 | 0.596 | 0.742 | 0.708 | 0.679 |
| SG | SVM | 0.708 | 0.636 | 0.742 | 0.596 | 0.742 | 0.708 | 0.679 |
| MSC | SVM | 0.738 | 0.455 | 0.667 | 0.541 | 0.667 | 0.738 | 0.624 |
| BSL | SVM | 0.754 | 0.477 | 0.681 | 0.568 | 0.681 | 0.754 | 0.642 |
| DERIV1 | SVM | 0.877 | 0.318 | 0.655 | 0.636 | 0.655 | 0.877 | 0.651 |
| DERIV2 | SVM | 0.831 | 0.500 | 0.711 | 0.667 | 0.711 | 0.831 | 0.697 |
| Raw+scaled | PLS | 0.738 | 0.477 | 0.676 | 0.553 | 0.676 | 0.738 | 0.633 |
| SG+scaled | PLS | 0.738 | 0.477 | 0.676 | 0.553 | 0.676 | 0.738 | 0.633 |
| MSC+scaled | PLS | 0.738 | 0.568 | 0.716 | 0.595 | 0.716 | 0.738 | 0.670 |
| BSL+scaled | PLS | 0.723 | 0.500 | 0.681 | 0.550 | 0.681 | 0.723 | 0.633 |
| DERIV1+scaled | PLS | 0.846 | 0.545 | 0.733 | 0.706 | 0.733 | 0.846 | 0.725 |
| DERIV2+scaled | PLS | 0.831 | 0.705 | 0.806 | 0.738 | 0.806 | 0.831 | 0.780 |
| Raw+scaled | SVM | 0.800 | 0.909 | 0.929 | 0.755 | 0.929 | 0.800 | 0.844 |
| SG+scaled | SVM | 0.800 | 0.909 | 0.929 | 0.755 | 0.929 | 0.800 | 0.844 |
| MSC+scaled | SVM | 0.815 | 0.773 | 0.841 | 0.739 | 0.841 | 0.815 | 0.798 |
| BSL+scaled | SVM | 0.862 | 0.795 | 0.862 | 0.795 | 0.862 | 0.862 | 0.835 |
| DERIV1+scaled | SVM | 0.938 | 0.727 | 0.836 | 0.889 | 0.836 | 0.938 | 0.853 |
| DERIV2+scaled | SVM | 0.892 | 0.705 | 0.817 | 0.816 | 0.817 | 0.892 | 0.817 |

**Notes.**

[a] Raw-with no pretreatment, SG- Savitsky-Golay smoothing, MSC- multiplicative scatter correction, BSL-baseline removal, DERIV1-the first derivative, DERIV2- the second derivative.

[b] Positive prediction value (PPV) = (true positive)/(true positive + false positive).

[c] Negative prediction value (NPV) = (true negative)/(true negative + false negative).

### Modeling importance of NIRS feature in SVM model

SVM-RFE algorithm ranked 1,557 NIRS features, which was shown in File S4. By progressively modeling with a combination of top 262 features in SVM, the overall prediction accuracy reached its max value of 0.862 (Fig. 6A). Besides, SVM model had its max sensitivity of 0.908 with only top 1 feature, and had its max specificity of 0.909 with top 220 features (Figs. 6B, 6C). Among the top 262 NIRS feature, 201 were in Band II (8,500 ~5,500 cm$^{-1}$), which referred to 1st overtone and their combinations of CH2/CH3/OH/NH stretching), eight were in Band I (12,500 ~8,500 cm$^{-1}$), which referred to 2nd overtone and their combinations of CH2/CH3/OH/NH stretching, and 53 were in Band III (5,500 ~4,000 cm$^{-1}$), which referred to combinations of CH2/CH3/OH/NH stretching and 2nd overtone of C =O stretching (File S5).

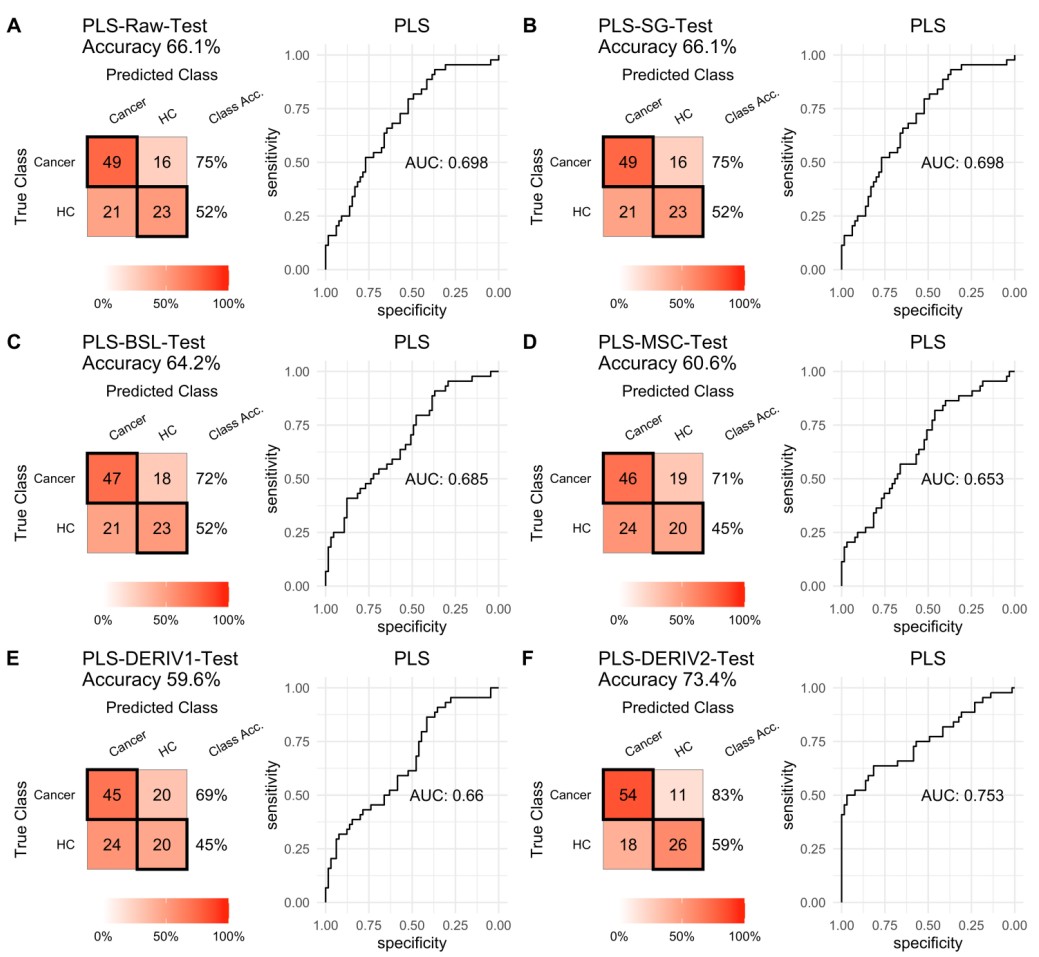

**Figure 2 PLS test.** Prediction performance of PLS models in testing with different pretreatment methods, including (A) raw NIRS, (B) SG: Savitzky-Golay smoothing, (C) BSL: baseline removal, (D) MSC: multiplicative scatter correction, (E) DERIV1: the first derivative, (F) DERIV2: the second derivative using confusion matrix tables and receiver operating characteristic (ROC) curves.

# DISCUSSION

Although NIRS has a wide range of applications in the medical field, such as brain function monitoring (*Grossmann, 2008*; *Holper et al., 2019*), muscle oxygenation monitoring (*Klusiewicz et al., 2021*), cardiac function monitoring (*Ortega-Loubon et al., 2019*), neonatal care (*Tran et al., 2021*), there are rare reports on liquid biopsy- based NIRS for cancer diagnosis. Our previous studies (*Zhu et al., 2023*; *Chen et al., 2021*) showed a promising diagnostic value in cancer using plasma or pleural effusion. Again, this pilot study revealed that urine-based had a very good prediction value for cancers.

## Advantages of urine based NIRS analysis for cancer diagnosis

Compared to the traditional cancer diagnosis methods, such as carcinoembryonic antigen test, CT/MRI scans, and pathology, which would cause pain or harm to some extent, urine based NIRS analysis has obvious advantages of noninvasiveness, convenience, and low

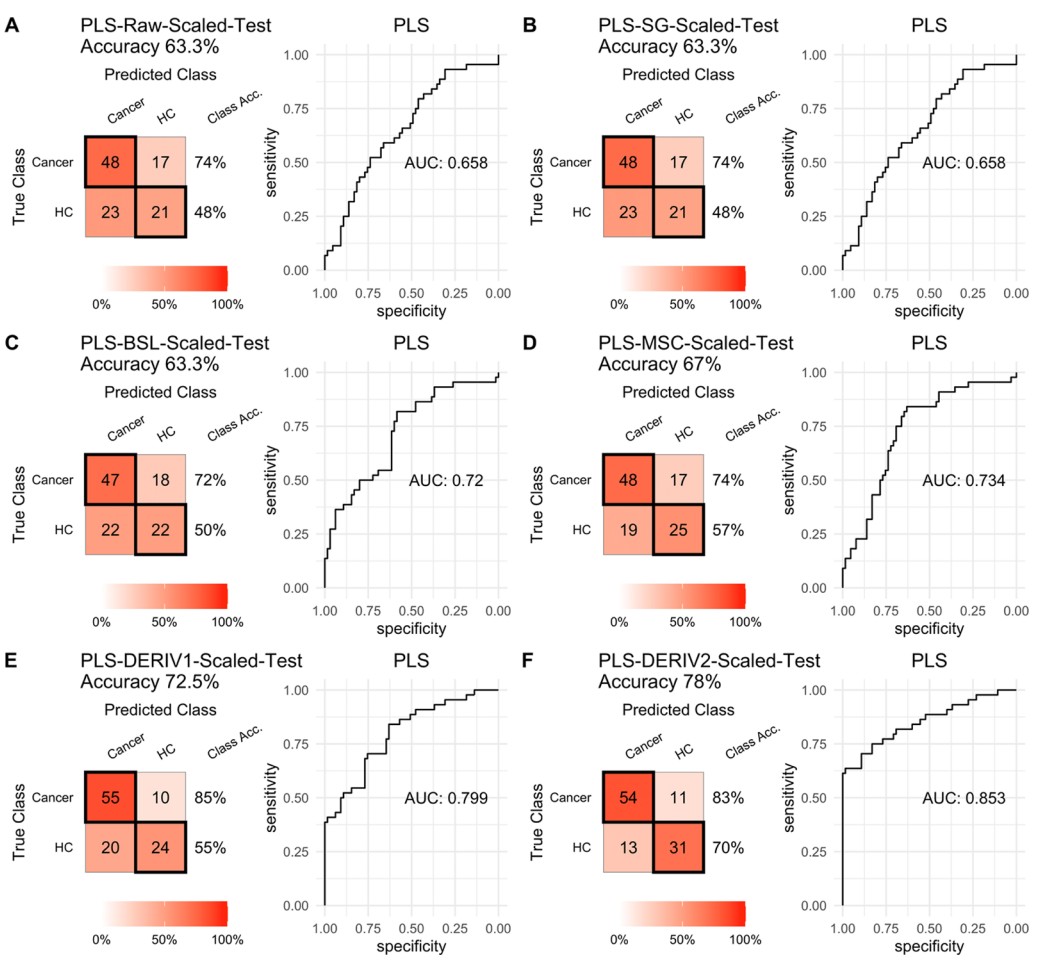

**Figure 3** **PLS test scaled.** Prediction performance of PLS models in testing with combined with "scaling" and "center" and different pretreatment methods, including (A) raw NIRS, (B) SG: Savitzky-Golay smoothing, (C) BSL: baseline removal, (D) MSC: multiplicative scatter correction, (E) DERIV1: the first derivative, (F) DERIV2: the second derivative using confusion matrix tables and receiver operating characteristic (ROC) curves.

cost. Therefore, urine based NIRS has a great potential as a cancer diagnosis application, especially in a daily monitoring way. Though LC-MS based omics are widely used in cancer biomarker discover studies, the equipment and analysis cost are expensive, and professional and technical personnel are required to perform the analysis. In contrast, NIRS equipment and its analysis cost is much lower than LC-MS, and it needs only a very simple sample preparation or even without preparation. Because NIRS equipment is relatively small, cheap, and portable, a further picture that this pilot study hints at are that there will be NIRS detectors in the home for daily monitoring for healthy status, for example, a "smart toilet" with a NIRS detector. Given these advantages of urine based NIRS diagnostics, further large-scale multicenter studies are encouraged.

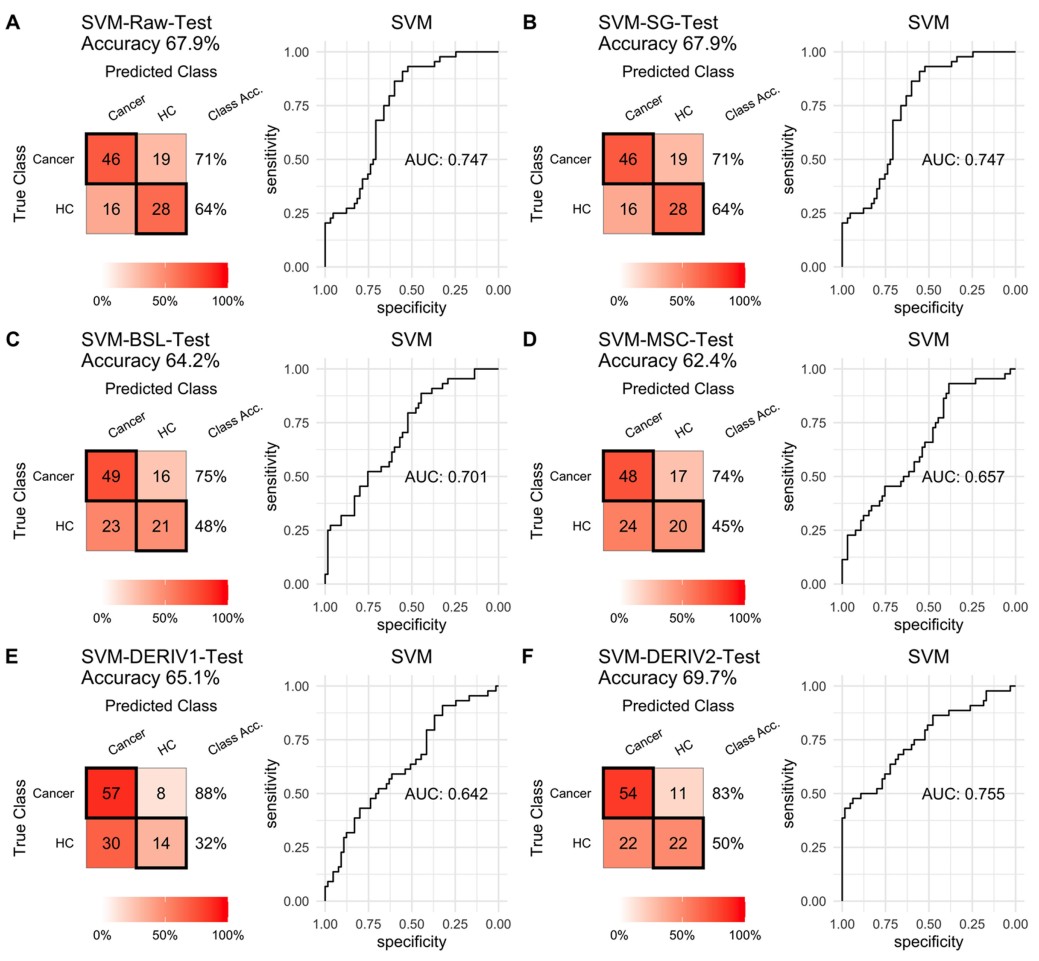

**Figure 4  SVM test.** Prediction performance of SVM models in testing with different pretreatment methods, including (A) Raw NIRS, (B) SG: Savitzky-Golay smoothing, (C) BSL: baseline removal, (D) MSC: multiplicative scatter correction, (E) DERIV1: the first derivative, (F) DERIV2: the second derivative using confusion matrix tables and receiver operating characteristic (ROC) curves.

## Machine learning algorithms

Unlike genomics, proteomics, and metabolomics, which can identify molecules in biological samples, NIRS provides comprehensive indicators of the functional groups of molecules. Machine learning algorithms are necessary to be combined in NIRS data analysis for diagnosis. There are a variety of machine learning algorithms available for modeling with NIRS data. PLS is one of the most commonly used algorithms in NIRS field due to its advantages as follows: (1) PLS is suitable for situations where the number of samples is less than the number of variables; (2) PLS provides interpretable results; (3) PLS can handle multicollinearity. While PLS is a linear model, and it was not always the best algorithm for complicated relationship between variable and interests. SVM algorithm (*Normanno et al., 2018*) could be used where the relationship between spectra and interest is presented as a non-linear. In this project, though PLS took only about half of the computational time of SVM, the prediction accuracy was much lower than that of SVM. In consistent with this,

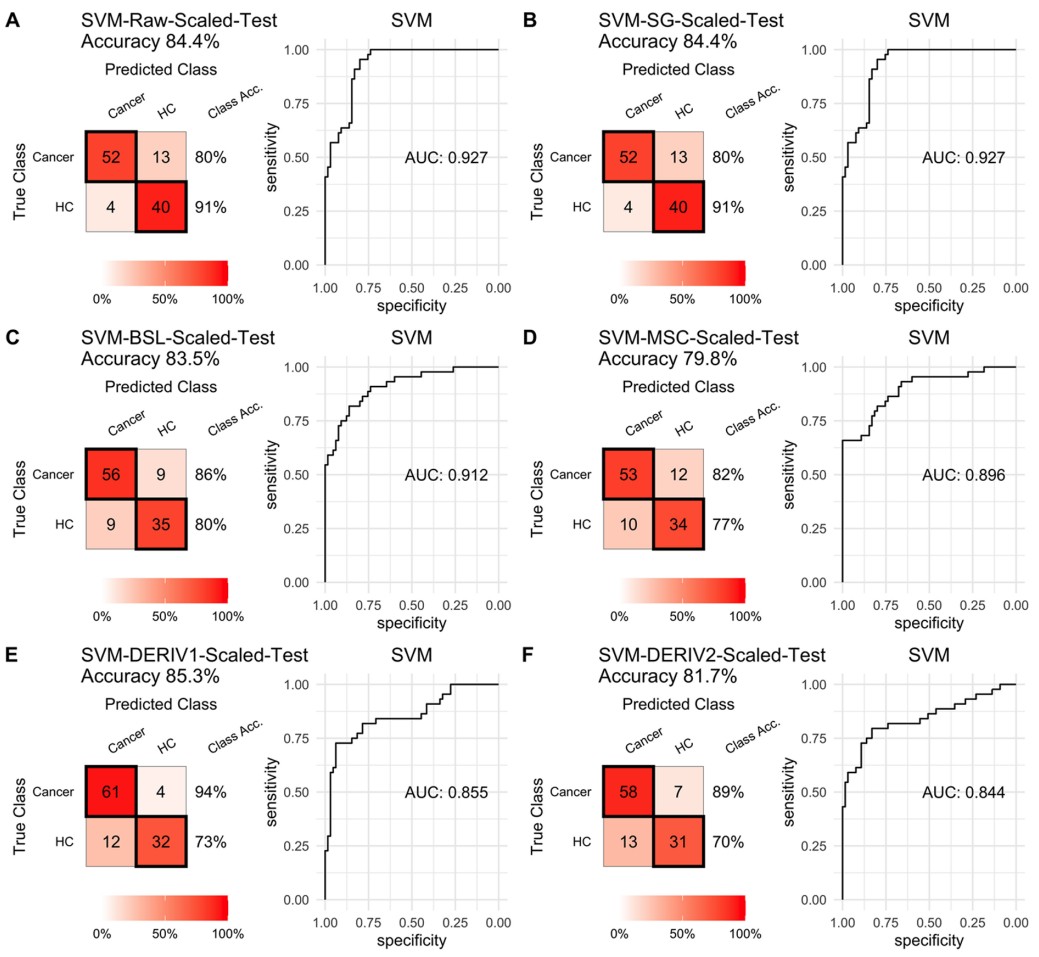

**Figure 5  SVM test scaled.** Prediction performance of SVM models in testing with combined with "scaling" and "center" and different pretreatment methods, including (A) raw NIRS, (B) SG: Savitzky-Golay smoothing, (C) BSL: baseline removal, (D) MSC: multiplicative scatter correction, (E) DERIV1: the first derivative, (F) DERIV2: the second derivative using confusion matrix tables and receiver operating characteristic (ROC) curves.

our previous NIRS based cancer diagnosis projects demonstrated SVM was more suitable than PLS in modeling with NIRS data from liquid biopsy samples.

Pretreatment is very important in NIRS data modeling, and this study illustrated that different models need their specific pretreatment methods, it is crucial to do a proper investigation in pretreatments. This project showed that only the second derivative improved the PLS, while the "scaling" and "center" pretreatment significantly improved the SVM model. Other pretreatments, including SG smoothing, multiplicative scatter correction, baseline removal, the first derivative, played a limited role in model performance, and even decreased the prediction accuracy in specific model, such as SVM modelling with the first derivative pretreated NIRS data. Therefore, it is highly recommended to carefully select pretreatment for the specific algorithm.

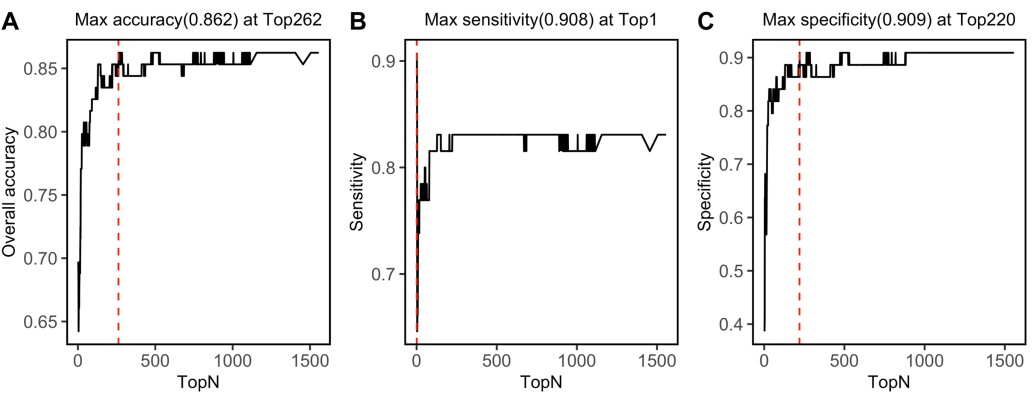

**Figure 6** **Top feature.** Prediction performance, including (A) accuracy, (B) sensitivity, and (C) specificity in testing set was investigated with modeling with progressively. Red dashed lines refer to the Top N when max values were reached.

## Assignment of the NIRS features

Though it is difficult to assign these NIRS features directly to specific molecules, due to weak absorbance, wide peak width, and overlapping of the NIRS peaks, NIRS can still provide a lot of useful information about the functional groups of the samples. There are three main NIRS bands, including Band I, Band II, and Band III, which cover a variety of functional groups of biological molecules, including metabolites, DNA, and proteins. In this study, among the top 262 NIRS features of SVM model, 201 were in Band II, which contained water spectral peaks. In the past, the main disadvantage of NIRS in liquid biopsy analysis was from influence from water spectral peaks in this range. Since Dr. Roumiana Tsenkova established a theory- "Aquaphotomics", in which water band was the main study object and the compounds in water interact with the OH of water, and then change the water spectral pattern (*Tsenkova, Kovacs & Kubota, 2015*; *Tsenkova et al., 2018*). With the inspiration of "Aquaphotomics", water status might be changed due to changes in concentrations of compounds in urine, including metabolites, DNA and proteins, which are widely reported to associated with diseases by metabolomics, proteomics, and genomics. Indeed, we tried modeling with a specific water spectral area, but we failed to obtain a better prediction outcome than that with the whole spectra (prediction accuracy in testing: 0.761). However, it is worthy to investigate the potential diagnostic role of "Aquaphotomics" in urine-based NIRS diagnosis. However, combinations with other analytical tools, such as LC-MS and NMR, are still needed to identify the specific biomarkers for diseases.

## Limitation is sample size

This is only a pilot study, the limitations in the following aspects must be listed. First, the sample size is relatively small, and urine samples were from only one center. Thus, our established urine based NIRS diagnosis method should undergo further validation with a large sample size study, especially containing validation samples from other centers. Second, this pilot study had too many types in a small size cancer group, and it might lead to a bigger intragroup variance and then a lower prediction accuracy. Thus, further study

with a proper number of cancer types from some specific system, and each cancer type has adequate cases for modeling. Third, the algorithms applied in this project were only simple ones, while more complicated model, such as deep learning algorithm, has not been used due to the limited sample size.

# CONCLUSIONS

This pilot study revealed an accurate, convenient, noninvasive, and low-cost cancer diagnosis method using combination urine based NIRS analysis and SVM modeling. Though the prediction accuracy in this study was not as high as those in our previous plasma or pleural effusion based NIRS diagnosis (*Zhu et al., 2023*; *Chen et al., 2021*), the highest prediction accuracy of 0.862 encouraged further studies with a large-scale sample from multi-centers to validate the application value of this method.

# ACKNOWLEDGEMENTS

We appreciate the help in NIRS equipment and data analysis from Professor Yongjiang Wu, Zhejiang University.

## Funding
This study was financially supported by the National Natural Science Foundation of China (81302840), the Zhejiang Natural Science Foundation (LY23H010002), and the Zhejiang Medical and Health Science Project (2022KY622&2022KY693). There was no additional external funding received for this study. The funders had no role in study design, data collection and analysis, decision to publish, or preparation of the manuscript.

## Grant Disclosures
The following grant information was disclosed by the authors:
National Natural Science Foundation of China: 81302840.
Zhejiang Natural Science Foundation: LY23H010002.
Zhejiang Medical and Health Science Project: 2022KY622&2022KY693.

## Competing Interests
The authors declare there are no competing interests.

## Author Contributions
- Jing Zhu performed the experiments, analyzed the data, authored or reviewed drafts of the article, and approved the final draft.
- Siyu Zhang performed the experiments, analyzed the data, authored or reviewed drafts of the article, and approved the final draft.
- Ruting Wang analyzed the data, prepared figures and/or tables, authored or reviewed drafts of the article, and approved the final draft.

- Ruhua Fang performed the experiments, authored or reviewed drafts of the article, and approved the final draft.
- Lan Lei analyzed the data, authored or reviewed drafts of the article, and approved the final draft.
- Ji Zheng conceived and designed the experiments, authored or reviewed drafts of the article, and approved the final draft.
- Zhongjian Chen conceived and designed the experiments, analyzed the data, authored or reviewed drafts of the article, and approved the final draft.

## Human Ethics

The following information was supplied relating to ethical approvals (i.e., approving body and any reference numbers):

Zhejiang Cancer Hospital granted Ethical approval to carry out the study within its facilities (IRB-2023-375).

## Data Availability

Raw data are available in the Supplemental Files.

## Supplemental Information

Supplemental information for this article can be found online at http://dx.doi.org/10.7717/peerj.15895#supplemental-information.

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
