# Peer review of "Urine based near-infrared spectroscopy analysis reveals a noninvasive and convenient diagnosis method for cancers: a pilot study"

_PeerJ, doi:10.7717/peerj.15895_

## Round 0.1 · original submission · Minor Revisions

Please address the concerns of both reviewers and amend the manuscript accordingly.

Reviewer 1 ·

Basic reporting

The article mentions the use of the near-infrared spectroscopy technique using urine as a sample from healthy and cancer patients for the identification of cancer cells.
English used in the manuscript was very professional.
Literature references were relevant but need to be put in the journal format.

The figures, and raw data which were shared were relevant and justified.

Experimental design

The primary aim of this research was to come up with a noninvasive and convenient diagnosis method for detecting cancers patient with the use of near-infrared spectroscopy.

The research question was very well defined as it's a pilot study, a lot of variables need to be experimented with and validated before applying it.

The method defined is well described and the authors have come up with an SVM module using a second derivative which is very specific for finding the changes in healthy and cancer patients.

The class of cancer patient chosen as Calss-3, need to understand the specificity for Class-1 and class-2 patients as well. Since it's a pilot study, hoping more experiments would be carried out to validate these findings.

Validity of the findings

The author had used Near Infrared spectroscopy in their earlier research articles, which uses blood samples using ATR-FTIR spectroscopy.
The research focuses on the simpler method to identify the cancer patients for which they have tested about 327 patients.
All the data which was analyzed have been shared and they have used different processing software and they have come up with different models and predication were carried out.

The findings still need to be screened on more combinations and come up with validated results that can be applied in real scenarios.

Additional comments

There are some corrections to be done which are listed below.

1. Line 40 – Spelling to be corrected – “randomly spitted into” to “ randomly split into”
2. Line-53 - Spelling to be corrected – “when top 262” to “ when the top 262”
3. Line-89 – Correction needed – “urine-based researches for biomarkers” to “urine-based research for biomarkers”
4. Line- 137 – “(PLS) was trained” to be corrected as “(PLS) were trained”
5. Line-148 – “according” to be corrected as “according to “
6. Line-155 – “spectrum was consisting” to be written S” spectrum consisted”
7. Line-183 – “term” to be written as “terms”
8. Line-187 – “able reached its” to be written as” able to reach its”
9. Line-188 – “was main factor” to be corrected as “was the main factor”
10. Line-194 – “which was showed” to be corrected as “which was shown”
11. Line-198-202 – Pl. correct the functional group details which are shown in Supplementary file – 5.
• The Vibration mode for band-II (8500 ~ 5500 cm-1) stretching corresponds to - CH2/CH3/OH/NH/SH
• The Vibration mode for Band I (12500 ~ 8500 cm-1) stretching corresponds to - CH2/CH3/OH/NH/SH
• The Vibration mode for Band III (5500 ~ 4000 cm-1) stretching corresponds to - CH2/CH3/OH/NH/C=O
• Also, can you pl. recheck the stretching’s of SH functional group – is it really between 8500-5500 cm-1 ?
12. Line- 224 – “at is that” to be corrected as “at are that”
13. Line-229 – Use capital letter “machine”
14. Line-237 – Kindly recheck this sentence. Are we saying PLS is better than SVM ?
“In this project, though PLS took only about half of the computational time of SVM, the prediction time was much lower than that of SVM”
15. All the references listed and cited in the manuscript are not as per the PeerJ format. e.g. –“ Smith et al (J of Methodology, 2005, V3, pp 123)”

Reviewer 2 ·

Basic reporting

Cancer poses a significant global health challenge, and the author has studied near-infrared spectroscopy (NIRS) for urine analysis and used a machine learning algorithm for its accurate interpretation. Artificial intelligence and machine learning are now become predominant in medicine and are successfully applied to improving human health.
Overall, the research adds value to cancer research, and the report is well-written for publication.

Experimental design

The experiments section discusses in detail and in relation to the study topic, particularly the application of five machine learning methodologies.
The author used a balance of data points, including 181 cancer cases and 146 healthy people, and attempted to use cancer cases from various subgroups, such as lung, gastric, and breast cancer, while maintaining random data for machine learning.

1. In Figure-4
Please double-check the AUC values of E and F as I have noticed a significant difference in them.

2. In Figure-7 plot D, author can provide more discussion on boxplot -D (p = 0.96) as compare to other box plots having p < 0.1

3. Supplementary file – 5.
Please recheck the statement “The -SH functional group stretching is between 8500-5500 cm-1”

Validity of the findings

The outcome of the results was comparable with the earlier work published,
Previous research by the author shown a promising diagnostic value in cancer utilizing plasma or pleural effusion, and the current work gives the relevant cancer prediction. However, it was shown that the suggested SVM models might predict cancers.

Additional comments

The study is limited to 327 people and requires more data to validate the SVM model, similar to how comparing NIRS data with other analytical technique data, such as LC-MS, provides more confidence and validation for the usage of the NIRS technique.

---

## Round 0.2 · accepted · Accept

All issues pointed out by the reviewers were adequately addressed and the manuscript was amended accordingly. Therefore, the revised version is acceptable now.